# Outcomes of COVID-19 Patients Admitted to the Intermediate Respiratory Care Unit: Non-Invasive Respiratory Therapy in a Sequential Protocol

**DOI:** 10.3390/ijerph191710772

**Published:** 2022-08-29

**Authors:** Mercè Gasa, Yolanda Ruiz-Albert, Ana Cordoba-Izquierdo, Mikel Sarasate, Ester Cuevas, Guillermo Suarez-Cuartin, Lidia Méndez, Julio-César Alfaro-Álvarez, Joan Sabater-Riera, Xosé L. Pérez-Fernández, María Molina-Molina, Salud Santos

**Affiliations:** 1Respiratory Department, Bellvitge Biomedical Research Institute (IDIBELL), Bellvitge University Hospital, 08907 L’Hospitalet de Llobregat, Spain; yolanda.ruiz@bellvitgehospital.cat (Y.R.-A.); acordoba@bellvitgehospital.cat (A.C.-I.); msarasate@bellvitgehospital.cat (M.S.); ester.cuevas@bellvitgehospital.cat (E.C.); gsuarezc@bellvitgehospital.cat (G.S.-C.); lmendezm@bellvitgehospital.cat (L.M.); mariamolina@bellvitgehospital.cat (M.M.-M.); 2Department of Medicine, Campus Bellvitge, Universitat de Barcelona, 08907 L’Hospitalet de Llobregat, Spain; 3Respiratory Department, Viladecans Hospital, 08840 Viladecans, Spain; jucealal@yahoo.es; 4Critical Care Department, Bellvitge Biomedical Research Institute (IDIBELL), Bellvitge University Hospital, 08907 L’Hospitalet de Llobregat, Spain; jsabater@bellvitgehospital.cat (J.S.-R.); josep@bellvitgehospital.cat (X.L.P.-F.)

**Keywords:** COVID-19, hypoxic respiratory failure, intermediate respiratory care unit, non-invasive respiratory therapy, high-flow nasal cannula, non-invasive ventilation

## Abstract

**Highlights:**

**What are the main findings?**

**What is the implication of the main finding?**

**Abstract:**

The intermediate respiratory care units (IRCUs) have a pivotal role managing escalation and de-escalation between the general wards and the intensive care units (ICUs). Since the COVID-19 pandemic began, the early detection of patients that could improve on non-invasive respiratory therapies (NRTs) in IRCUs without invasive approaches is crucial to ensure proper medical management and optimize limiting ICU resources. The aim of this study was to assess factors associated with survival, ICU admission and intubation likelihood in COVID-19 patients admitted to IRCUs. Observational retrospective study in consecutive patients admitted to the IRCU of a tertiary hospital from March 2020 to April 2021. Inclusion criteria: hypoxemic respiratory failure (SpO_2_ ≤ 94% and/or respiratory rate ≥ 25 rpm with FiO_2_ > 50% supplementary oxygen) due to acute COVID-19 infection. Demographic, comorbidities, clinical and analytical data, and medical and NRT data were collected at IRCU admission. Multivariate logistic regression models assessed factors associated with survival, ICU admission, and intubation. From 679 patients, 79 patients (12%) had an order to not do intubation. From the remaining 600 (88%), 81% survived, 41% needed ICU admission and 37% required intubation. In the IRCU, 51% required non-invasive ventilation (NIV group) and 49% did not (non-NIV group). Older age and lack of corticosteroid treatment were associated with higher mortality and intubation risk in the scheme, which could be more beneficial in severe forms. Initial NIV does not always mean worse outcomes.

## 1. Background

COVID-19 is a complex multisystem disease with pulmonary involvement being the most prevalent manifestation. Respiratory features range from a mild disease in more than 80%, through moderate to severe hypoxic respiratory failure (HRF) in near 15%, and up to a critical disease in less than 5% of patients [1]. Providing optimal supplemental respiratory support and monitoring is crucial to maintain individualized target oxygen saturation by pulse oximetry (SpO_2_) while the patient is overcoming the disease. Before the COVID-19 outbreak, the benefits of intermediate respiratory care units (IRCUs) were well-documented. The IRCU is an area for monitoring and treating patients with acute or exacerbated respiratory failure caused by a disease that is primarily respiratory. The essential aim is adequate and appropriate cardiorespiratory monitoring and/or treatment of respiratory insufficiency by noninvasive techniques. The IRCU reduces intensive care unit (ICU) admission time, optimizes ICU bed capacity, and reduces mortality and health care costs [2]. At the beginning of the COVID-19 pandemic, there was a lack of evidence regarding the most effective respiratory management for this patient. Now, emerging data supports the application of non-invasive respiratory therapies (NRTs) including high flow nasal cannula (HFNC) and non-invasive ventilation (NIV) as cost-effective resources in many patients. Almost 19% of COVID-19 patients are successfully treated with NRTs [3,4].

Nowadays, the IRCU has a pivotal role to manage escalation and de-escalation between general wards and ICUs. Recognizing patients that will benefit more from NRTs in IRCUs without ICU transfer has become a crucial challenge to ensure optimal medical management and to make proper use of limiting resources. Thus, our objective was to analyze all patients admitted to our IRCU due to COVID-19-related HRF during the first year of the COVID-19 outbreak (before initiating population vaccination) and to assess factors associated with survival, ICU transfer, and intubation rates in the entire cohort and the according NRTs required.

## 2. Methods

Study design: An observational–retrospective study was conducted including patients admitted to the IRCU of a Tertiary Hospital in Barcelona (Spain) from March 2020 to April 2021. The final follow-up date was 28 June 2021. The study protocol was approved by the local ethics committee of the Hospital Universitari de Bellvitge (PR260/20).

Inclusion criteria: 1. Acute COVID-19 infection (positive polymerase chain reaction for SARS-CoV2 from nasopharyngeal swab at hospital admission); 2. Clinical signs of pneumonia (fever, cough, and dyspnea); 3. HRF defined by an oxygen saturation (SpO_2_) ≤ 94% and/or a respiratory rate (RR) ≥ 25 rpm with supplementary oxygen with an inspired oxygen fraction (FiO_2_) > 50%. These inclusion criteria concur with definitions for severe, mild, and moderate critical disease from the WHO living guidance for COVID-19 [5]. Patients were admitted to the IRCU from emergency departments or transferred from regional hospitals or general wards due to clinical impairment.

Exclusion criteria: (1) Criteria for direct ICU admission (imminent intubation, hemodynamic instability, multiorgan failure, abnormal mental status, and shock requiring support with vasoactive drugs).

In our institution, a local COVID multidisciplinary team was formed at the beginning and went through all periods of the pandemic. The main purpose of this team was to allocate scarce resources with priority for those with the highest probability of benefiting from them, relying on ethical principles based on objective and widely shared criteria, preventing arbitrary decisions, and guaranteeing equity [6,7]. This team included pulmonologists, intensivists, and other medical colleagues. For each patient, an agreed plan was stated based on comorbidities, baseline fragility, and severity of COVID-19 disease to provide the best medical care irrespective of treatment effort.

Patient clinical and laboratory data were collected from electronic medical records. Data regarding hospital stays included total and relative hospital lengths and admission setting (general ward, IRCU or advanced IRCU if ICU transfer was needed in less than 24 h). Data regarding NRT: NRT type (oxygen reservoir mask, HFNC, HFNC followed by NIV, and initial NIV); time on each NRT (HFNC, intermittent and continuous NIV), maximal FiO_2_; NIV parameters (inspiratory positive airway pressure (IPAP), expiratory positive airway pressure (EPAP). The escalation/de-escalation algorithm for sequential NRT is shown in Figure 1.

NIV was indicated if the optimal respiratory target (SpO_2_ > 94% and/or RR < 25 breaths/minute) was not achieved after 1 h on the HFNC trial and then, intensivists were advised for a possible intubation if NIV failed during the next 24–36 h. Furthermore, NIV was also considered if the pulmonologist had a high clinical suspicion of ventilatory failure despite HRF (PaCO_2_ > 45 mmHg) and/or concomitant sleep respiratory disorder, especially if obesity coexisted. Commonly, NIV therapy was started with an EPAP from 8 to 12 cmH_2_O and IPAP from 12 to 16 cmH_2_O with little modifications to improve patient tolerance/comfort. Depending on clinical status, NIV was maintained or disrupted for small periods to allow oral intake and family–social communication.

Medical data included tocilizumab and remdesivir. Systemic corticosteroid data were collected in four categories based on the RECOVERY trial [8] published within the period of the present study: (1) no corticosteroid; (2) HIT (intravenous bolus of 1–2 mg/Kg/day methylprednisolone or its equivalent dexamethasone dose for 3 days or dexamethasone 6 mg/day orally or intravenously for 10 days; (3) HIT + TAP (option 2 followed by oral prednisone starting from 0.5 mg/Kg/day, tapering the dose over 7 to 10 days; (4) TAP (option 3 without previous bolus). All participants were treated according to current hospital protocols.

The decision to transfer a patient to the ICU was assessed and reassessed as needed by the multidisciplinary COVID team. Patients were categorized depending on survival status, ICU transfer, and intubation rate.

Statistical analyses: Data were expressed as mean ± standard deviation(SD) for continuous data and frequency(percentage) for categorical data. Bivariate comparisons were evaluated using Chi-squared (categorical), student’s T (parametric) or Mann–Whitney (nonparametric) unpaired tests. Multiple comparisons were evaluated using Chi-squared(categorical), student’s T (parametric) and Mann–Whitney tests, applying the Bonferroni method when significant differences were found by the Kruskal–Wallis test (nonparametric). The relationship between dependent variables (survival, intubation, or ICU transfer) and independent factors (variables being statistical and/or clinically significant in the bivariate comparison analysis) was evaluated by logistic regression analysis. The association results were summarized using unadjusted and adjusted odds ratios and β coefficients with their 95% confidence intervals. A *p*-value < 0.05 was considered statistically significant. SPSS version 22 software (SPSS Inc., Chicago, IL, USA) was used for all the analyses.

## 3. Results

Six-hundred seventy-nine patients were admitted to the IRCU during the study period (Figure 2).

Seventy-nine patients had orders to not intubate. The remaining 600 patients were included in further analysis. Mean age was 61 ± 11 years and 33% were female. All patients were admitted during the pre-vaccine period, 42% during the first wave (from March-10 2020 until 31 August 2020), 24% during the second wave (from 1 September 2020 until 31 December 2020), and 34% during the third wave (1 January 2021 until 30 June 2021). A total of 44% were admitted firstly to the general ward and 56% directly to the IRCU. SaFiO_2_ and PaFiO_2_ mean ratios at IRCU admission were 158 ± 68 and 156 ± 83. A total of 13% required Monaghan, 33% HFNC, 41% first trial of HFNC followed by NIV, and 8% initial NIV. A total of 88% received corticosteroids (53% bolus plus tapering), 41% tocilizumab and 17% remdesivir. A total of 57% (344 patients) improved without ICU transfer; 43% (256 patients) required ICU admission; 36% (220 patients) required intubation and 19% (116 patients) died.

### 3.1. Characterization Depending on Severity: Moderate vs. Severe HRF

At IRCU admission, 51% (307 patients) presented severe HRF (SaFiO_2_ mean ratio of 139 ± 51 requiring NIV) and 49% (293 patients) presented moderate HRF (mean SaFiO_2_ mean ratio of 173 ± 76 with no NIV requirement). Table 1 compares patients regarding NIV needs.

The NIV group was slightly older (63 ± 11 vs. 59 ± 11 years, *p* < 0.001) and had more chronic renal failure (12% vs. 6%, *p* 0.026) compared with the non-NIV group. At IRCU admission, the NIV group had higher RR (24 ± 6 vs. 23 ± 5 breaths/minute) and worse inflammatory profile than the non-NIV group. Average length of hospital stay was longer in the NIV group than in the non-NIV group (35 ± 31 vs. 20 ± 19 days, *p* < 0.001) due to longer post-IRCU stay but similar pre-IRCU and IRCU stays. Patients requiring NIV received similar tocilizumab and remdesivir medications but higher and larger doses of corticosteroids. The NIV group had higher ICU transfer and intubation rates (69% vs. 16% *p* < 0.001 and 57% vs. 15% *p* < 0.001, respectively) with a significantly greater mortality rate (33% vs. 5%, *p* < 0.001).

### 3.2. Mortality, ICU Transfer, and Intubation Rates in NIV Group (Severe HRF)

From this subgroup (307 patients), 84% (257 patients) received an HFNC trial before initiating NIV and 16% (50 patients) started initial NIV. A total of 31% (95 patients) improved with no need for ICU admission (Table 2). The remaining 69% (212 patients) required ICU transfer: a total of 176 patients (83%) required intubation and 102 patients finally died (48% of those requiring ICU and 33% of the entire subgroup).

Non-survivors were older, presented chronic renal failure and were on chronic immunosuppressant therapies more frequently. At IRCU admission, non-survivors had similar respiratory load (SaFiO_2_, PaFiO_2_ and RR), CRP and LDH levels but higher D-dimer values. Percentage of patients requiring initial NIV or NIV after HFNC trial did not differ between groups. Patients that finally died were on continuous NIV more days and received less corticosteroids, mainly bolus and tapering scheme. Intubated patients were older and less obese compared with non-intubated patients. At IRCU admission, patients that finally were intubated had worse inflammatory profile but similar SaFiO_2_. Intubated patients were on HFNC and on intermittent NIV for less time and received less corticosteroids (specifically bolus + tapering scheme) and remdesivir than non-intubated. A total of 50% of patients requiring intubation finally died (88 patients of 176 patients). By contrast, in non-intubated patients, 27% (36 of 132 patients) required ICU admission and only 11% (14 patients) finally died.

### 3.3. Mortality, ICU Transfer and Intubation Rates in Non-NIV Group (Moderate HRF)

From this subgroup (293 patients), 73% (213 patients) received HFNC and 27% (80 patients) received an oxygen reservoir mask during IRCU stay. A total of 83% (243 patients) improved with no need for ICU admission (Table 3). The remaining 16% (46 patients) were transferred to the ICU: 96% (44 patients) required intubation and finally 10 patients died (22% of those requiring ICU and 5% of the entire subgroup).

Non-survivors compared to survivors were older, had more previous cardiopathy and were admitted more frequently in the first wave. At IRCU admission, non-survivors had higher levels of CRP but similar SaFiO_2_. Maximal FiO_2_ and total time on HFNC therapy was similar in both groups. Non-survivors received less corticosteroids than alive patients. ICU transfer and intubation rates were significantly different between groups (dead vs. alive: 71% vs. 13% and 71% vs. 12%, respectively). Intubated patients had longer pre-IRCU stays and were admitted in more proportion during the first wave compared with non-intubated patients. They had higher LDH and CRP levels at IRCU admission and remained for less time on HFNC receiving higher maximal FiO_2_ but similar corticosteroid regimens.

### 3.4. Predicting Factors of Patient Outcomes: Survival, ICU Transfer, and Intubation Rates

Table 4 summarized outcomes based on NRT during IRCU stay.

Logistic regression analysis determines factors associated with higher survival, intubation, and ICU transfer. In the global cohort (600 patients), independent factors of survival were younger age, less chronic immunosuppressive therapy, and receiving corticosteroids (bolus plus tapering regimen increased 11-fold) based on the multivariate model adjusted for intubation rate, SaFiO_2_ at IRCU admission, and wave admission at hospital (Figure 3).

Factors associated with survival, ICU transfer, and intubation in the entire cohort were: IRCU: intermediate respiratory care unit; OT: oral intubation; H + T: bolus and progressive tapering; NIV: non-invasive ventilation; SaFiO_2_: oxygen saturation by pulse oximetry divided by inspired oxygen fraction; DD: D-dimer; LDH: lactate dehydrogenase; PaFiO_2_: partial arterial pressure of oxygen divided by inspired oxygen fraction; HFNC: high flow nasal cannula; Y/N: yes/no.

Adding cardiopathy, D-dimer >3000, and higher LDH did not change the results. Factors associated with ICU transfer were NIV requirement and higher levels of CRP and LDH at IRCU admission. No association with age or corticosteroids were found. Factors associated with intubation were older age and not receiving corticosteroids (specifically bolus plus tapering scheme). In multivariate models, tocilizumab or remdesivir lost their significance.

For severe HRF patients (NIV group), having levels of d-dimer ≥ 3000 μg/L at IRCU admission in the multivariate model adjusted for many cofactors was also associated with higher mortality (Figure 4).

Multivariate logistic regression analysis factors associated with survival, ICU transfer, and intubation in the NIV group were: IRCU: intermediate respiratory care unit; NIV: non-invasive ventilation; H/T: hit and progressive tapering; SaFiO_2_: oxygen saturation by pulse oximetry divided by inspired oxygen fraction; HFNC: high flow nasal cannula; CRP: C-reactive protein; LDH: lactate dehydrogenase.

Factors associated with ICU transfer were D-dimer level ≥ 3000 and higher LDH levels at IRCU admission after adjustments. Adding obesity and hepatopathy to the model did not change the results. Factors associated with intubation were higher age and CRP at IRCU admission; taking corticosteroids (any regimen) and less IRCU stay were associated with less intubation risk.

For moderate HRF patients (non-NIV group), factors associated with survival were younger age, receiving corticosteroids (any scheme), and obviously not needing intubation (Figure 5).

Multivariate logistic regression analysis factors associated with survival probability, ICU transfer, and intubation risks in the non-NIV group were: IRCU: intermediate respiratory care unit; NIV: non-invasive ventilation; Y/N: yes/no; H/T: hit and progressive tapering; SaFiO_2_: oxygen saturation by pulse oximetry divided by inspired oxygen fraction; HFNC: high flow nasal cannula; CRP: C-reactive protein; LDH: lactate dehydrogenase; NIV: non-invasive ventilation; IRCU: intermediate respiratory care unit; SaFiO_2_: oxygen saturation by pulse oximetry divided by inspired oxygen fraction; Y/N: yes/no; H/T: hit/tap; LDH: lactate dehydrogenase; CRP: C-reactive protein.

Presenting higher LDH levels at IRCU admission was the unique factor associated with more ICU transfer. Intubation was associated with longer pre-IRCU stay, being admitted at first wave, higher LDH and CRP levels, and receiving tocilizumab. By contrast no effect of age was found.

## 4. Discussion

The present work shows that in COVID-related HRF, 57% of patients are optimally treated at the IRCU with no need for ICU admission. This is highly noted in patients with moderate HRF, where 84% improved mainly on HFNC and only 16% required ICU transfer for advanced invasive therapies; but also in those with severe HRF requiring NIV, where 67% improved at the IRCU without ICU transfer. This highlights the pivotal role of the IRCU over the course of one year of the COVID pandemic.

To achieve the best potential benefit of any NRT on HRF management, the patient should be allocated to a monitored setting, being cared for by personnel experienced in NRT. The IRCU is a suitable setting for this purpose where qualified respiratory staff can manage this situation with efficiency and less cost than the ICU [9,10]. Nevertheless, ensuring prompt endotracheal intubation is mandatory if the patient presents acute deterioration or no improvement after an NRT short trial. In our institution, a good coordination/agreement between pulmonologists and intensivists has been crucial to handle this situation. The implementation of a sequential NRT protocol at the IRCU has allowed treating adequately many patients suffering from moderate–severe HRF, providing the opportunity to safeguard intensive care capacity for those requiring invasive therapies. The impact of NRTs on HRF is still controversial among non-COVID [11] and COVID patients [12]. From a recent systematic review and meta-analysis [11] including 3804 patients with HRF (PaO_2_/FiO_2_ ratio < 200, immunocompromised included), NIV via a face mask lowered the risk of both intubation and mortality, and HFNC lowered only intubation compared with conventional oxygen. Regarding COVID-19, Crimi C. et al. [13] found that among 364 randomized patients with COVID- 19 pneumonia and mild hypoxemia (PaFiO_2_ < 300 but ≥ 200), the use of HFNC did not significantly reduce the likelihood of escalation of respiratory support compared with standard oxygen. However, this finding could not be extrapolated to our cohort due to a severe grade of hypoxemia (see Table 1, PaFiO_2_ mean ratio of 156). At present, there have been three randomized controlled trials [14,15,16] that examined NRT’s impact on HRF due to COVID-19 with severe hypoxemia. Compared therapies, primary outcomes, and study settings differed among these trials, making it difficult to extract overall conclusions. The HiFLo-Covid trial [14] reported a lower risk of intubation and time to clinical recovery with HFNC compared with conventional oxygen in COVID-19 patients admitted to ICU (median PaO_2_/FiO_2_ ratio < 100 at randomization). The HENIVOT trial [15] did not find differences for the primary outcome (days free of respiratory support) between Helmet-NIV and HFNC in COVID-19 patients in an ICU setting (median PaO_2_/FiO_2_ ratio ≈ 100 at randomization) although a lower percentage of patients required intubation in the Helmet-NIV group. In the RECOVERY-RS trial [16] conducted in hospitalized COVID-19 patients, CPAP lowers tracheal intubation or mortality within 30 days compared with standard oxygen, but no benefit was found with HFNC compared with standard oxygen. The median PaO_2_/FiO_2_ ratio at randomization was ≈ 115 for all three groups. There is no NIV group (pressure support) designed for comparison between CPAP and HFNC groups. Despite limitations because of the observational design, in the present study, initial NIV did not always imply worse prognosis. From those receiving initial NIV, 68% of patients survived and 38% survived with no need for ICU transfer.

The prompt recognition of rapidly worsening despite maximal NRT escalation is critical to improve survival and, in this situation, intubation should not be delayed. In our cohort, worse outcomes were found in those that started on HFNC at IRCU admission but during the following 36 h required progressive escalation until maximal NRT (continuous NIV FiO_2_ 100%) to maintain optimal respiratory targets (SpO_2_ and RR). From our data, we could not find any parameters at IRCU admission that help us to recognize the likelihood of worsening. Moreover, remaining longer in the general ward before IRCU transfer could be wrongly categorized as less severe than reliable (concept of silent hypoxia) [17,18,19]. At IRCU admission, these subjects got worse than expected and quickly deteriorated. Considering the current available evidence [12,20,21] of NRT’s impact on COVID-related HRF and waiting for more on-going RCT [22,23], we suggest not to delay intubation in patients that experience quick respiratory worsening during the first 24–36 h of IRCU admission and in patients being on continuous NIV for 48 h or more.

In line with previous randomized controlled trials conducted in COVID-19 hospitalized patients in general settings [8] and especially in ICU settings, [24,25] best outcomes are linked to systemic corticosteroid therapy. Furthermore, we found that a bolus plus progressive tapering scheme could be more beneficial than only a bolus scheme in specific patients, probably in more severe cases. However, from the present work, this affirmation cannot be certain since it was not designed for this purpose and the benefits and risks of taking corticosteroids during medium–large periods were not analyzed. Moreover, it seems that the inflammatory profile at IRCU admission is linked to worse outcomes as pointed out in many previous studies [26,27]; we observed that specifically higher levels of D-dimer [6,28] and CRP [29,30] are the most relevant biomarkers with potential usefulness as biomarkers of COVID-19 severity, although their efficacy in predicting treatment response is still inconclusive [31,32].

The main limitation of our study is its observational design that does not allow analyzing cause–effect relationships. Furthermore, some patients included mostly during the first month of the pandemic presented missing data (such as PaFiO_2_ and RR at IRCU admission). However, the present work includes a quite wide cohort of severe COVID patients treated by the same pulmonologists and intensivists through this entire period in a tertiary hospital with well-standardized protocols.

As future directions to guide research, it is necessary to obtain competent evidence about NRT’s impact on COVID-related HRF. More well-designed randomized controlled trials are mandatory to answer the main question: when not to delay intubation if is necessary and conversely, when to avoid intubation if it is not necessary. On-going observational data are also important to assess the real effect of vaccination; we should be aware that despite complete vaccination, many patients will still not be protected against COVID-19 (especially those immunocompromised and a few presumably immunocompetent).

## 5. Conclusions

The 57% of patients suffering from COVID-related HRF are well-treated at the IRCU with no need for ICU admission; mainly in those requiring HFNC but also in many requiring NIV. Higher survival and lower risk of intubation/ICU transfer are related with systemic corticosteroid therapy. A hit and tapering scheme could be more beneficial than only a hit bolus in severe patients. A rapid respiratory worsening despite maximal NIT involves no intubation delay but starting NIV does not always mean worse outcomes.

## Figures and Tables

**Figure 1 ijerph-19-10772-f001:**
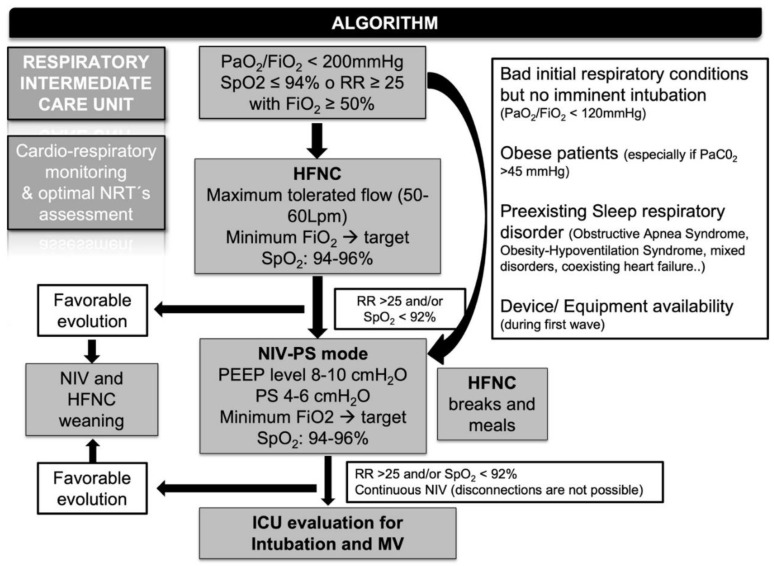
Algorithm at IRCU: Sequential non-invasive respiratory support; sequential non-invasive respiratory support. NRT, non-invasive respiratory therapy; IRCU, intermediate respiratory care unit; PaFiO_2_, partial arterial pressure of oxygen divided by inspired oxygen fraction; SpO_2_, oxygen saturation by pulse oximetry; RR, respiratory rate; FiO_2_, inspired oxygen fraction; HFNC, high flow nasal cannula; NIV, non-invasive ventilation. PEEP; positive end-expiratory pressure; PS, pressure support; PaCO_2_: partial arterial pressure of carbon dioxide; MV, mechanical ventilation.

**Figure 2 ijerph-19-10772-f002:**
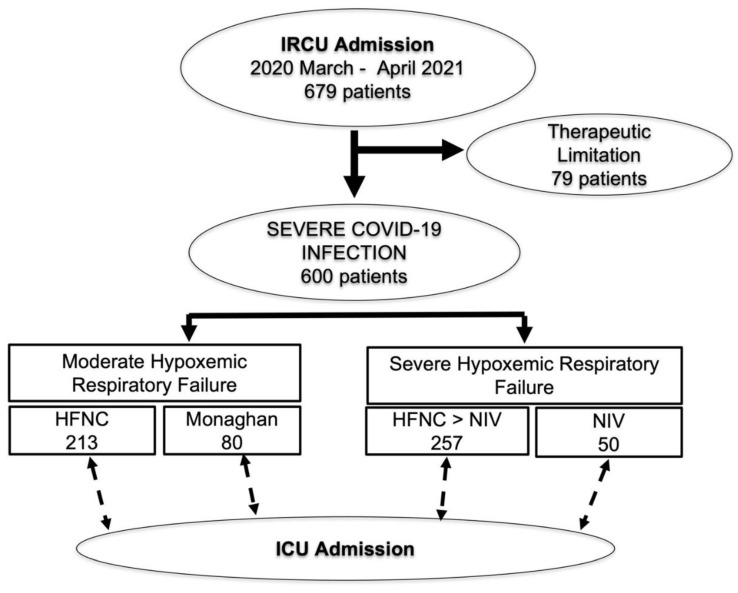
Patient flowchart. Flowchart of patients admitted to IRCU during the study period. IRCU, Intermediate respiratory care unit; HFNC, high flow nasal cannula; NIV, non-invasive ventilation; ICU, intensive care unit.

**Figure 3 ijerph-19-10772-f003:**
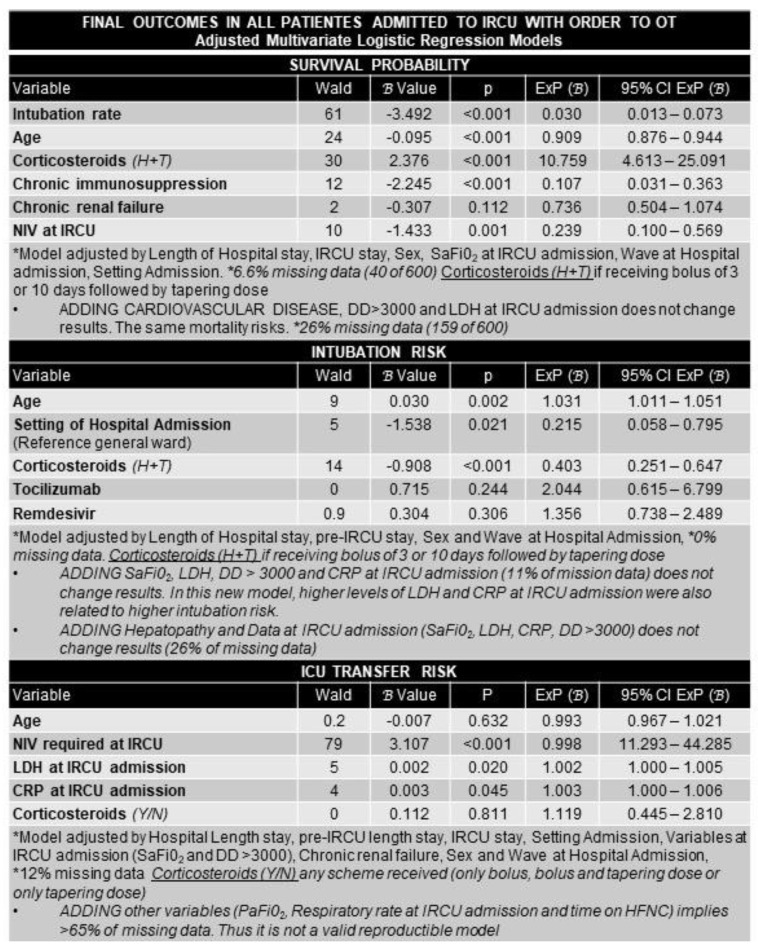
Multivariate logistic regression analysis. Factors associated with survival, ICU transfer, and intubation in the entire cohort.

**Figure 4 ijerph-19-10772-f004:**
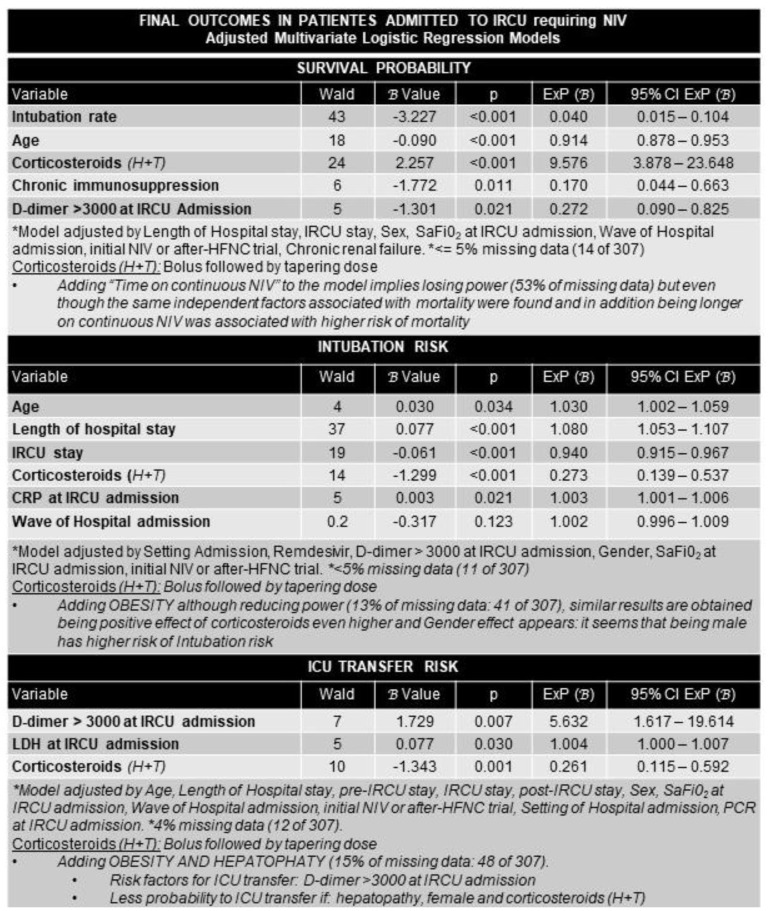
Multivariate logistic regression analysis. Factors associated with survival, ICU transfer, and intubation in the NIV group.

**Figure 5 ijerph-19-10772-f005:**
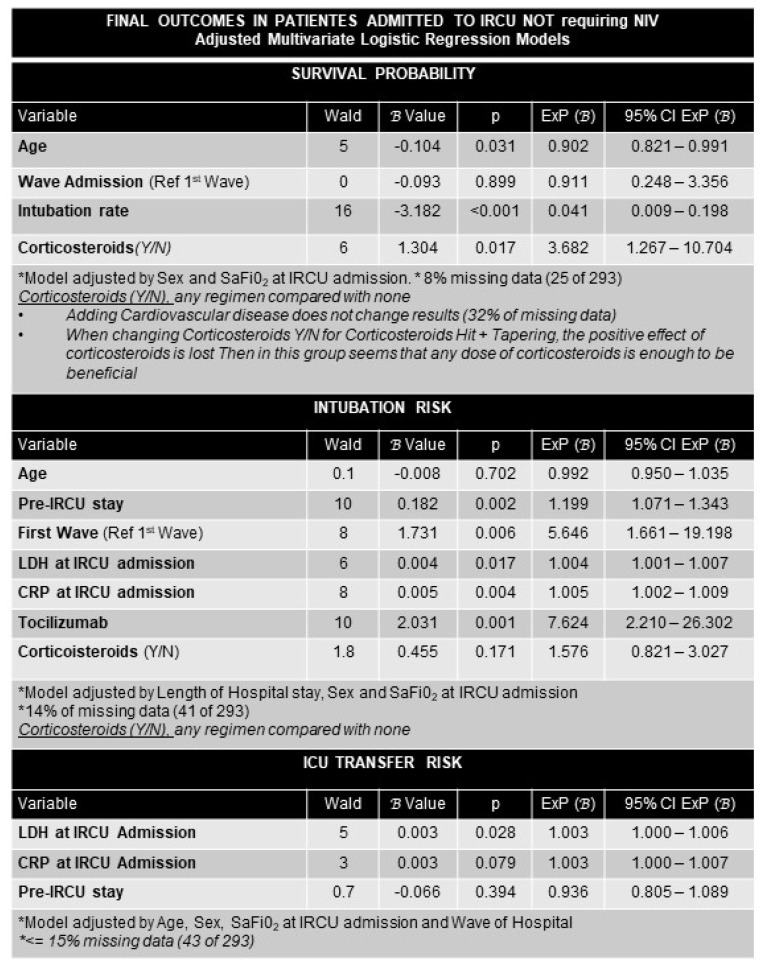
Multivariate logistic regression analysis. Factors associated with survival, ICU transfer, and intubation in the non-NIV group.

**Table 1 ijerph-19-10772-t001:** Characteristics of the entire cohort and depending on NIV requirements. NIV, non-invasive ventilation; SaFiO_2_, oxygen saturation by pulse oximetry divided by inspired oxygen fraction; PaFiO_2_, partial arterial pressure of oxygen divided by inspired oxygen fraction; HTA, arterial systemic hypertension; OSA, obstructive sleep apnea; COPD, chronic obstructive pulmonary disease; IS therapy, immunosuppressive therapy; IRCU, intermediate respiratory care unit; ICU, intensive care unit; OT, oral intubation; RR, respiratory rate; PaCO_2_, partial arterial pressure of carbon dioxide; LDH, lactate dehydrogenase; NRT, non-invasive respiratory therapy; HFNC, high flow nasal cannula; FiO_2_, inspired oxygen fraction; H + T, bolus and progressive tapering.

	TOTAL *n* = 600	NIV Not Required *n* = 293	NIV Required *n* = 307	*p*
SaFiO_2_	158.0 (68)	173 (76)	139 (51)	<0.001
PaFiO_2_	156.4 (83)	184 (93) *n* = 123	137 (68) *n* = 172	<0.001
Age (years)	61 (11)	59 (11)	63 (11)	<0.001
Female (*n*, %)	193 (32%)	96 (33%)	97 (31%)	NS
HTA (*n*, %)	276 (48%)	116 (49%)	160 (55%)	NS
Dyslipidemia (*n*, %)	242 (42%)	102 (45%)	142 (49%)	NS
Diabetes (*n*, %)	145(25%)	58 (27%)	87 (31%)	NS
Obesity (*n*, %)	176 (30%)	72 (32%)	104 (37%)	NS
Cardiovascular disease (*n*, %)	71 (12%)	27 (13%)	44 (16%)	NS
Respiratory disease (*n*, %) None OSA COPD Asthma	493 (82%) 53(10%) 26 (4%) 24 (4%)	243 (83%) 24 (8%) 12 (4%) 12 (4%)	250 (81%) 29 (9%) 14 (4%) 12 (4%)	NS
Chronic kidney failure (*n*, %)	56 (9%)	18 (6%)	38 (12%)	0.026
History of malignancy (*n*, %)	71 (12%)	37 (17%)	34 (12%)	NS
Chronic liver disease (*n*, %)	42 (7%)	22 (10%)	20 (7%)	NS
Chronic IS therapy (*n*, %)	30 (5%)	10 (3%)	20 (7%)	NS
Length of stay (days)	29.5 (30.0)	20.1 (19.2)	34.7 (31.0)	<0.001
Length pre-IRCU stay (days)	4.8 (13.5)	3.0 (6.1)	2.5 (6.5)	NS
Length of IRCU stay (days)	8.7 (12.2)	7.9 (9.1)	8.6 (13.2)	NS
Length post-IRCU stay (days)	14.5 (24.4)	9.3 (16.3)	24.0 (30.9)	<0.001
Wave of Hospital Admission (*n*, %) 1st (March 20–August 20) 2nd (Sep 20–Dec 20) 3rd (Jan 2021–June 21)	256 (42%) 144 (24%) 202 (34%)	136 (46%) 74 (25%) 83 (28%)	120 (39%) 70 (23%) 119 (38%)	0.029
Setting of Hospital admission (*n*, %) General Ward IRCU Advanced IRCU	274 (44%) 307 (50%) 21 (6%)	162 (55%) 125 (43%) 6 (20%)	112 (36%) 182 (59%) 15 (5%)	<0.001
Length IRCU Admission—OT (days)	5.7 (4.5)	4.8 (6.9) *n* = 29	5.9 (4.0) *n* = 165	NS
Length ICU admission—OT (days)	1.7 (2.9)	1.4 (2.4)	1.8 (3.0)	NS
RR (breaths/minute)	23.9 (5.5)	23.2 (4.9) *n* = 147	24.4 (5.9) *n* = 190	0.045
PaCO_2_ (mmHg)	36.3 (9.7)	35.3 (5.4) *n* = 124	37.0 (11.9) *n* = 173	NS
Seric Bicarbonate (mEq/L)	28.4 (26.3)	29.3 (31.7)	27.8 (21.7)	NS
Ferritin (ng/L)	1745 (2690)	1753 (2854)	1764 (2629)	NS
LDH (U/L)	434 (181)	392 (134)	472 (203)	<0.001
D-dimer (mcg/L)	1831 (5897)	1137 (3147)	2386 (7459)	0.014
C-reactive protein (mg/L)	135 (146)	123 (106)	151 (175)	0.019
NRT in IRCU (*n*, %) Monaghan HFNC HFNC→NIV Initial NIV	80 (13%) 213 (33%) 257 (41%) 50 (8%)	80 (27%) 213 (73%) - -	- - 257 (84%) 50 (16%)	-
FiO_2_ HFNC (%)	90.3 (50.8)	80.7 (13.4) *n* = 135	97.6 (6.5) *n* = 175	0.003
Time on HFNC (days)	4.0 (3.8)	6.0 (3.5) *n* = 135	2.6 (3.3) *n* = 188	<0.001
Corticosteroids (*n*, %) No treatment Bolus Bolus + tapering Low-dose	75 (12%) 193 (32%) 316 (53%) 19 (3%)	50 (17%) 83 (28%) 150 (51%) 10 (3%)	25 (8%) 110 (36%) 166 (54%) 9 (3%)	0.005
Tocilizumab (*n*, %)	242 (41%)	108 (37%)	134 (43%)	0.068
Remdesivir (*n*, %)	93 (17%)	52 (18%)	41 (13%)	NS
ICU transfer rate (*n*,%)	258 (43%)	46 (16%)	212 (69%)	<0.001
Intubation rate (*n*, %)	220 (36%)	44 (15%)	176 (57%)	<0.001
Survival rate (*n*, %)	469 (78%)	279 (95%)	205 (67%)	<0.001

**Table 2 ijerph-19-10772-t002:** Comparison depending on survival and intubation status in the NIV group: patients with severe hypoxemic respiratory. OT, oral intubation; HTA, arterial systemic hypertension; OSA, obstructive sleep apnea; COPD, chronic obstructive pulmonary disease; IS therapy, immunosuppressive therapy; IRCU, intermediate respiratory care unit; ICU, intensive care unit; SaFiO_2_, oxygen saturation by pulse oximetry divided by inspired oxygen fraction; PaFiO_2_, partial arterial pressure of oxygen divided by inspired oxygen fraction; LDH, lactate dehydrogenase; RR, respiratory rate; PaCO_2_: partial arterial pressure of carbon dioxide; NRT, non-invasive respiratory therapy; HFNC, high flow nasal cannula; NIV, non-invasive ventilation; FiO_2_, inspired oxygen fraction; IPAP, inspiratory positive airway pressure; EPAP, expiratory positive airway pressure; H + T, bolus and progressive tapering.

	NIV Required *n* = 307	DEAD *n* = 102	ALIVE *n* = 205	*p*	OT *n* = 176	NO OT *n* = 132	*p*
Age (years)	63 (11)	68 (8)	60 (11)	<0.001	64 (11)	61 (11)	0.008
Female (*n*, %)	97 (31%)	37 (36%)	60 (29%)	NS	59 (33%)	38 (29%)	NS
HTA (*n*, %)	160 (55%)	55 (55%)	105 (55%)	NS	90 (53%)	70 (58%)	NS
Dyslipidemia (*n*, %)	142 (49%)	51 (50%)	89 (48%)	NS	88 (51%)	53 (46%)	NS
Diabetes (*n*, %)	87 (31%)	26 (26%)	61 (33%)	NS	54 (32%)	33 (29%)	NS
Obesity (*n*, %)	104 (37%)	29 (30%)	74 (41%)	0.076	53 (31%)	50 (45%)	0.026
Cardiovascular disease (*n*, %)	44 (16%)	20 (20%)	23 (13%)	NS	22 (13%)	21 (19%)	NS
Respiratory disease (*n*, %) None OSA COPD Asthma	250 (81%) 29 (9%) 14 (4%) 12 (4%)	76 (74%) 11 (11%) 8 (8%) 4 (45)	172 (84%) 18 (9%) 6 (3%) 8 (4%)	NS	146 (83%) 12 (7%) 9 (5%) 6 (3%)	103 (78%) 17 (13%) 5 (4%) 6 (4%)	NS
Chronic kidney failure (*n*, %)	38 (12%)	25 (24%)	13 (6%)	<0.001	24 (14%)	15 (12%)	NS
History of malignancy (*n*, %)	34 (12%)	14 (14%)	20 (11%)	NS	24 (14%)	10 (9%)	NS
Chronic liver disease (*n*, %)	20 (7%)	4 (4%)	16 (9%)	NS	9 (5%)	11 (10%)	NS
Chronic IS therapy (*n*, %)	20 (7%)	13 (13%)	7 (3%)	0.014	13 (7%)	7 (5%)	NS
Length of hospital stay (days)	34.7 (31.0)	28.3 (22.4)	37.4 (33.4)	0.013	44.0 (36.9)	22.5 (12.8)	<0.001
Length of pre-IRCU stay (days)	2.5 (6.5)	2.0 (5.1)	2.8 (7.1)	NS	2.7 (7.6)	2.3 (4.6)	NS
Length of IRCU stay (days)	8.6 (13.2)	6.0 (4.8)	9.9 (15.7)	0.003	7.3 (15.1)	10.2 (10.0)	0.048
Length of post-IRCU stay (days)	24.0 (30.9)	20.3 (22.1)	24.8 (33.7)	NS	34.0 (36.9)	10.0 (10.1)	<0.001
Wave at H. admission (*n*, %) 1st (March 20–August 20) 2nd (September 20–December 20) 3rd (January 21–June 21)	120 (39%) 70 (23%) 119 (38%)	36 (35%) 34 (33%) 32 (31%)	84 (41%) 36 (18%) 85 (41%)	0.007	77 (44%) 39 (22%) 60 (34%)	43 (33%) 32 (23%) 58 (44%)	NS
Setting at H. admission (*n*, %) General Ward IRCU Advanced IRCU	112 (36%) 182 (59%) 15 (5%)	38 (37%) 63 (62%) 1 (1%)	74 (36%) 117 (57%) 14 (7%)	0.080	53 (30%) 113 (64%) 10 (6%)	59 (45%) 86 (51%) 5 (4%)	0.030
ICU transfer (*n*, %)	210 (69%)	96 (94%)	114 (57%)	<0.001	176 (100%)	36 (27%)	<0.001
Intubation rate (*n*, %)	175 (57%)	89 (87%)	86 (42%)	<0.001	176 (100%)	-	-
Length IRCU Adm.—OT (days)	5.9 (4.0) *n* = 165	7.2 (4.1) *n* = 89	4.3 (3.3) *n* = 75	<0.001	5.9 (4.0) *n* = 165	5.9 (4.0) *n* = 165	-
Length ICU Adm.—OT (days)	1.8 (3.0)	2.1 (3.1)	1.4 (3.0)	NS	1.8 (3.0)	1.8 (3.0)	-
SaFiO_2_	139 (51)	135 (47)	140 (53)	NS	138 (51)	140 (51)	NS
PaFiO_2_	137 (68) *n* = 172	135 (64) *n* = 58	138 (71) *n* = 124	NS	126 (55) *n* = 92	138 (51) *n* = 79	0.021
RR (breaths/minute)	24.4 (5.9) *n* = 190	24.7 (5.7) *n* = 64	24.4 (5.9) *n* = 124	NS	25.0 (5.9) *n* = 100	23.9 (5.7) *n* = 90	NS
PaCO_2_ (mmHg)	37.0 (11.9) *n* = 173	36.4 (10.6) *n* = 58	37.4 (12.6) *n* = 113	NS	35.6 (8.4) *n* = 92	38.6 (14.8) *n* = 80	NS
Seric Bicarbonate (mEq/L)	27.8 (21.7)	25.1 (6.8)	27.5 (19.3)	NS	28.9 (19.1)	26.4 (6.3)	NS
Ferritin (ng/L)	1764 (2629)	1704 (1709)	1599 (1672)	NS	1748 (2957)	1772 (2173)	NS
LDH (U/L)	472 (203)	500 (211)	457 (198)	NS	501 (217)	433 (178)	0.004
D-dimer (mcg/L)	2386 (7459)	4130 (10,801)	1540 (4920)	0.004	2999 (8637)	1601 (5550)	0.009
C-reactive protein (mg/L)	151 (175)	147 (115)	148 (183)	NS	178 (212)	116 (99)	0.002
NRT in IRCU (*n*, %) HFNC -> NIV Initial NIV	257 (84%) 50 (16%)	86 (84%) 16 (16%)	171 (83%) 34 (17%)	NS	145 (84%) 31 (18%)	113 (86%) 19 (14%)	NS
FiO_2_ HFNC (%)	97.6 (6.5) *n* = 175	93.2 (4.5) *n* = 60	98.0 (8.1) *n* = 113	NS	98.8 (9.1) *n* = 89	91.2 (8.4) *n* = 85	NS
FiO_2_ NIV (%)	94.3 (12.1) *n* = 181	97.4 (7.9) *n* = 64	92.5 (13.8) *n* = 115	0.009	97.7 (6.5) *n* = 96	90.3 (15.5) *n* = 84	<0.001
IPAP (cmH_2_O)	14.5 (1.9) *n* = 190	15.0 (2.0) *n* = 65	14.2 (1.8) *n* = 120	0.008	14.4 (1.8) *n* = 100	14.7 (2.0) *n* = 86	NS
EPAP (cmH_2_O)	8.4 (1.5) *n* = 190	8.4 (1.6) *n* = 65	8.4 (1.5) *n* = 123	NS	8.4 (1.4) *n* = 100	8.4 (1.7) *n* = 89	NS
Time on HFNC (days)	2.6 (3.3) *n* = 188	2.2 (2.8) *n* = 64	3.0 (3.6) *n* = 122	NS	1.8 (2.6) *n* = 98	3.6 (3.8) *n* = 89	<0.001
Time on intermittent NIV (days)	2.7 (3.2) *n* = 189	2.1 (2.3) *n* = 62	3.0 (3.6) *n* = 125	NS	1.6 (1.7) *n* = 97	3.8 (4.0) *n* = 91	<0.001
Time on continuous NIV (days)	1.6 (2.7) *n* = 189	2.5 (3.9) *n* = 60	1.1 (1.2) *n* = 95	0.002	1.5 (1.3) *n* = 92	1.8 (3.9) *n* = 64	NS
Corticosteroids (*n*, %) No treatment Bolus Bolus +Tapering Tapering	25 (8%) 110 (36%) 166 (54%) 9 (3%)	5 (5%) 65 (64%) 27 (26%) 5 (5%)	20 (10%) 44 (21%) 138 (67%) 3 (2%)	<0.001	16 (9%) 76 (43%) 78 (44%) 6 (3%)	9 (7%) 33 (25%) 88 (67%) 2 (1%)	0.001
Corticosteroids (H + T) *n*, %	166 (54%)	27 (26%)	138 (67%)	<0.001	78 (44%)	88 (67%)	<0.001
Tocilizumab (*n*, %)	134 (43%)	49 (48%)	95 (46%)	NS	68 (39%)	65 (49%)	NS
Remdesivir (*n*, %)	41 (13%)	11 (11%)	30 (15%)	NS	16 (9%)	25 (19%)	0.012
ICU transfer rate (*n*, %)	210 (69%)	96 (94%)	114 (57%)	<0.001	176 (100%)	36 (27%)	<0.001
Intubation rate (*n*, %)	175 (57%)	89 (87%)	86 (42%)	<0.001	176 (100%)	-	-
Survival rate (*n*, %)	205 (67%)	-	-	-	86 (49%)	119 (90%)	<0.001

**Table 3 ijerph-19-10772-t003:** Characteristics depending on survival and intubation status in the non-NIV group: patients with moderate hypoxemic respiratory. OT, oral intubation; HTA, arterial systemic hypertension; OSA, obstructive sleep apnea; COPD chronic obstructive pulmonary disease; IS therapy, immunosuppressive therapy; IRCU, intermediate respiratory care unit; ICU, intensive care unit; SaFiO_2_, oxygen saturation by pulse oximetry divided by inspired oxygen fraction; PaFiO_2_, partial arterial pressure of oxygen divided by inspired oxygen fraction; LDH, lactate dehydrogenase; RR, respiratory rate; PaCO_2_: partial arterial pressure of carbon dioxide; NRT, non-invasive respiratory therapy; HFNC, high flow nasal cannula; NIV, non-invasive ventilation; FiO_2_, inspired oxygen fraction; IPAP, inspiratory positive airway pressure; EPAP, expiratory positive airway pressure; H + T, bolus and progressive tapering.

	NIV Not Required *n* = 293	DEAD *n* = 14	ALIVE *n* = 279	*p*	OT *n* = 44	No OT *n* = 249	*p*
Age (years)	59 (11)	65 (6)	59 (11)	0.027	60 (10)	59 (11)	NS
Female (*n*, %)	96 (33%)	3 (21%)	93 (33%)	NS	9 (20%)	87 (35%)	0.059
HTA (*n*, %)	116 (49%)	8 (57%	108 (48%)	NS	16 (42%)	100 (50%)	NS
Dyslipidemia (*n*, %)	102 (45%)	9 (64%)	94 (44%)	NS	20 (51%)	82 (44%)	NS
Diabetes (*n*, %)	58 (27%)	5 (36%)	53 (26%)	NS	9 (24%)	49 (27%)	NS
Obesity (*n*, %)	72 (32%)	5 (36%)	67 (32%)	NS	13 (34%)	59 (32%)	NS
Cardiovascular disease (*n*, %)	27 (13%)	4 (31%)	23 (12%)	0.048	6 (16%)	21 (12%)	NS
Respiratory disease (*n*, %) None OSA COPD Asthma	243 (83%) 24 (8%) 12 (4%) 12 (4%)	10 (71%) 2 (14%) 0% 2 (14%)	233 (83%) 24 (8%) 12 (4%) 10 (4%)	NS	34 (77%) 5 (11%) 1 (2%) 4 (9%)	209 (84%) 21 (8%) 11 (4%) 8 (3%)	NS
Chronic kidney failure (*n*, %)	18 (6%)	2 (15%)	16 (6%)	NS	2 (5%)	16 (6%)	NS
History of malignancy (*n*, %)	37 (17%)	4 (31%)	33 (16%)	NS	6 (16%)	31 (17%)	NS
Chronic liver disease (*n*, %)	22 (10%)	2 (15%)	20 (10%)	NS	2 (5%)	20 (11%)	NS
Chronic IS therapy (*n*, %)	10 (3%)	1 (7%)	9 (3%)	NS	0%	10 (4%)	NS
Length of hospital stay (days)	20.1 (19.2)	22.1 (17.6)	20.0 (19.3)	NS	39.2 (27.5)	16.7 (15.1)	<0.001
Length of pre-IRCU stay (days)	3.0 (6.1)	2.5 (3.5)	3.1 (6.3)	NS	9.2 (12.5)	1.9 (3.0)	<0.001
Length of IRCU stay (days)	7.9 (9.1)	3.8 (3.3)	8.1 (9.3)	NS	8.1 (14.8)	7.9 (7.8)	NS
Length of post-IRCU stay (days)	9.3 (16.3)	15.8 (18.9)	9.0 (16.2)	NS	21.9 (26.7)	7.1 (12.6)	<0.001
Wave at H. admission (*n*, %) 1st (March 20–August 20) 2nd (September 20–December 20) 3rd (January 21–June 21)	136 (46%) 74 (25%) 83 (28%)	9 (64%) 5 (36%) 0%	127 (45%) 69 (25%) 83 (30%)	0.055	28 (64%) 11 (25%) 5 (11%)	108 (43%) 63 (25%) 78 (31%)	0.014
Setting at H. admission (*n*, %) General Ward IRCU Advanced IRCU	162 (55%) 125 (43%) 6 (20%)	7 (50%) 7 (50%) 0%	155 (56%) 118 (42%) 6 (2%)	NS	25 (57%) 13 (29%) 6 (14%)	137 (55%) 112 (45%) 0%	<0.001
ICU transfer (*n*, %)	46 (16%)	10 (71%)	36 (13%)	<0.001	38 (86%)	8 (3%)	<0.001
Intubation rate (*n*, %)	44 (15%)	10 (71%)	34 (12%)	<0.001	44 (100%)	-	-
Length IRCU Adm.—OT (days)	4.8 (6.9) *n* = 29	5.9 (10.7) *n* = 10	4.2 (4.0) *n* = 19	NS	4.8 (6.9) *n* = 29	-	-
Length ICU Adm.—OT (days)	1.4 (2.4) *n* = 29	1.6 (3.3) *n* = 10	1.3 (1.7) *n* = 19	NS	1.4 (2.4)	-	-
SaFiO_2_	173 (76)	171 (103) *n* = 12	173 (75) *n* = 257	NS	171 (66)	173 (77)	NS
PaFiO_2_	184 (93) *n* = 123	105 (21) *n* = 3	186 (94) *n* = 120	0.004	171 (79) *n* = 9	185 (94) *n* = 115	NS
RR (breaths/minute)	23.2 (4.9)	24.7 (6.1) *n* = 3	23.2(4.9) *n* = 144	NS	25.1 (4.8) *n* = 9	23.1 (4.9) *n* = 138	NS
PaCO_2_ (mmHg)	35.3 (5.4) *n* = 124	34.3 (4.2) *n* = 3	35.3 (5.4) *n* =121	NS	36.5 (7.7) *n* = 9	35.2 (5.2) *n* = 115	NS
Seric Bicarbonate (mEq/L)	29.3 (31.7)	24.2 (2.8)	29.4 (32.1)	NS	25.2 (3.2) *n* = 9	29.6 (32.9)	NS
Ferritin (ng/L)	1753 (2854)	992 (967)	1784 (2904)	NS	1720 (1604)	1757 (2987)	NS
LDH (U/L)	392 (134)	429 (139)	391 (134)	NS	453 (135)	383 (132)	0.011
D-dimer (mcg/L)	1137 (3147)	740 (765)	1155 (3214)	NS	789 (1036)	1188 (3345)	NS
C-reactive protein (mg/L)	123 (106)	196 (135)	119 (103)	0.010	182 (137)	115 (98)	0.001
NRT in IRCU (*n*, %) Monaghan HFNC	80 (27%) 213 (73%)	4 (29%) 10 (71%)	76 (27%) 203 (73%)	NS	15 (34%) 29 (66%)	65 (26%) 184 (74%)	NS
FiO_2_ HFNC (%)	80.7 (13.4) *n* = 135	83.3 (16.1) *n* = 3	80.6 (13.4) *n* = 132	NS	95.0 (2.8) *n* = 9	79.9 (13.3) *n* = 128	0.001
Time on HFNC (days)	6.0 (3.5) *n* = 135	5.0 (6.1) *n* = 3	6.0 (3.5) *n* = 132	NS	2.4 (1.9) *n* = 9	6.1 (3.5) *n* = 128	0.006
Corticosteroids (*n*, %) None Bolus Bolus + Tapering Tapering	50 (17%) 83 (28%) 150 (51%) 10 (3%)	5 (36%) 8 (57%) 1 (7%) 0%	45 (16%) 75 (27%) 149 (53%) 10 (4%)	0.004	8 (18%) 14 (32%) 18 (41%) 4 (9%)	42 (17%) 69 (28%) 132 (53%) 6 (2%)	NS
Corticosteroids (H + T) *n*, %	150 (51%)	1 (7%)	149 (53%)	<0.001	18 (41%)	132 (53%)	NS
Tocilizumab (*n*, %)	108 (37%)	4 (29%)	104 (37%)	NS	9 (20%)	99 (40%)	0.050
Remdesivir (*n*, %)	52 (18%)	1 (7%)	51 (18%)	NS	6 (14%)	46 (18%)	NS
ICU transfer rate (*n*,%)	46 (16%)	10 (71%)	36 (13%)	<0.001	44 (100%)	2(<1%)	<0.001
Intubation rate (*n*, %)	44 (15%)	10 (71%)	34 (12%)	<0.001	-	-	-
Survival rate (*n*, %)	279 (95%)	-	-	-	34 (77%)	245 (98%)	-

**Table 4 ijerph-19-10772-t004:** Summarized outcomes based on NRT received during IRCU. ^a^ Monaghan group compared with the other groups. HFNC, high flow nasal cannula; NIV, non-invasive ventilation; ICU, intensive care unit.

Primary Outcomes	Monaghan *n* = 80	HFNC *n* = 213	HFNC→NIV *n* = 257	Initial NIV *n* = 50	*p*
**Age** (years)	60 (7)	60 (11)	63 (11)	65 (50)	0.007
**SaFiO_2_**	217 (63)	151 (37)	142 (46)	120 (38)	<0.001 ^a^
**ICU transfer rate** (*n*, %)	17 (21%)	32 (15%)	167 (64%)	31 (62%)	<0.001
**Intubation rate** (*n*, %)	15 (19%)	29 (14%)	145 (56%)	31 (62%)	<0.001
**Survival rate** (*n*, %) Global Not ICU-transfer ICU-transfer patients Intubated patients	76 (95%) 62 (99%) 14 (82%) 12 (80%)	203/213 (95%) 181 (98%) 22 (76%)	173 (64%) 78 (94%) 93 (54%) 69 (48%)	34 (68%) 13 (93%) 21 (58%) 18 (52%)	

## Data Availability

All data generated or analyzed during this study are included in this article. Further inquiries can be addressed to the corresponding author.

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
