# Peer review of "Outcomes of COVID-19 Patients Admitted to the Intermediate Respiratory Care Unit: Non-Invasive Respiratory Therapy in a Sequential Protocol"

_ijerph, 2022, doi:10.3390/ijerph191710772_

Round 1
Reviewer 1 Report
Thank you for asking me to review the paper entitled "Outcomes of COVID-19 Patients Admitted to the Intermediate Respiratory Care Unit: Non-Invasive Respiratory Therapy in a Sequencial Protocol".
It is interesting finding from "real life data analysis" of the outcomes of the treatment of COVID patients in the Intermediate Respiratory Care. Results are expected but some confirmation was needed.
I have minor suggestions:
Tables 1-4 have a black line instead of heading on the top. It is not clear which group is in which column.
Highlights: Line 16, page 1, the authors should add "in our patient population, the 57%.... at the beginning of the sentence.
Abstract: I would delete the first sentence and start with: The aim of this study was to assess ....
Line 36, older instead of longer age
Line 54, please add one sentence explaining what IRCU is to someone who does not know anything about it.
The rest is acceptable
Author Response
Dear Reviewer 1,
I appreciate all your comments. I have check and fit all them in order to improve the manuscript.
I have corrected the heading on the top of the Tables 1-4 and all the other changes you suggest.
Also in Line 54, I have added one first sentence explaining what IRCU is: “The IRCU is an area for monitoring and treating patients with acute or exacerbated respiratory failure caused by a disease that is primarily respiratory. The essential aim is adequate and appropriate cardiorespiratory monitoring and/or treatment of respiratory insufficiency by noninvasive techniques”.
Please see the attachment for the new version of the manuscript
Best regards
Dra Mercè Gasa
Reviewer 2 Report
I have read the paper presented by Mercè Gasa and colleagues about the outcomes of covid-19 patients admitted to the intermediate respiratory care unit. The study is overall well conducted. I have only few minor issues that need to be addressed before publication.
1- The abstract is poorly understandable. There are passages in which something is missing. I suggest reformulating these ones (eg " Recognizing patients that best benefited from non-invasive respiratory therapies 24 (NRTs) in intermediate respiratory care units (IRCU) is crucial to ensure management and limiting 25 resources during COVID pandemic. To assess factors associated with survival, intensive care unit 26 (ICU) admission and intubation likelihood in COVID-19 patients admitted to IRCU." Is "study aim" missing between these two sentences?
2- SpO2, FiO2, PaO2 and PaCO2 need to be checked throughout the text and figures. I have found several times that the "O" of "oxygen" in these acronyms is written ad a "0" "zero" (eg line 53: Sp02).
3- The headings of Tables 1 to 4 are missing.
4- I suggest adding three references:
a) https://thorax.bmj.com/content/early/2022/05/16/thoraxjnl-2022-218806
The authors need to check the results and the subsequent discussion in light of the recent RCT in which the effects of HFNO in COVID-19 patients with mild hypoxemia are minimal compared to conventional oxygen therapy, with regards to respiratory support escalation.
b) https://rc.rcjournal.com/content/67/2/227.short
c) Crimi C, Noto A, Cortegiani A, et al. Noninvasive respiratory support in acute hypoxemic respiratory failure associated with COVID-19 and other viral infections. Minerva Anestesiol. 2020;86(11):1190-1204. doi:10.23736/S0375-9393.20.14785-0
Extensive reviews on respiratory support in COVID-19 patients
Author Response
Dear Reviewer 2,
I appreciate your comments.
I have reformulated the abstract in order to make it understandable. A clear sentence has been add remarking the main objective of the study.
In addition, I have check all the acronyms of “oxygen” and I have written them in the right form (with a “O” instead “0, zero”). Thank you very much for this appreciation.
The headings of Table 1 to 4 are corrected.
Regarding the three references you mentioned, the first and the third references have been add in the “discussion”. For sure, they improve the discussion of this work. The second reference is not added because when I search it at pubmed is referred to a topic not related to COVID: “ Effect of Alirocumab on Lipoprotein(a) and cardiovascular risk after acute coronary syndrome”.
Thank you again for your great advices
Please see the attachment for the new version of the manuscript
Cordially,
Dr Mercè Gasa